# SYMMETRIC DUAL-PATH INTEGRATION FOR PROTEIN INVERSE FOLDING

## ABSTRACT

Protein inverse folding aims to recover amino acid sequences for a given 3D protein structure, underpinning broad applications such as enzyme engineering and drug discovery.Current methods often follow a serial pipeline, in which a structure encoder predicts a coarse sequence, which is then refined by protein language models (PLMs). However, because PLMs only perform post-hoc sequence edits, the refinement is bounded by the quality of upstream predictions.Thanks to recent multimodal protein language models (MPLMs), we could directly encode structure to generate sequences with pretrained structural knowledge, but we observe that they are not effective for inverse folding. Therefore, we introduce a harmonic dual-path architecture that both leverages PLMs for pretrained sequence knowledge and MPLMs for pretrained structural knowledge to iteratively guide protein sequence generation.Through extensive experiments across standard protein inverse folding benchmarks, our method achieves state-of-the-art performance, surpassing prior approaches, and ablation studies validate the rationale of our symmetric design, revealing a promising direction for the community.

## 1 INTRODUCTION

Protein sequence and structure are two sides of the same coin. The protein sequence specifies the order of amino acids (residues), and once known, the corresponding protein can be chemically synthesized (Defresne et al., 2021; Khakzad et al., 2023; Huang et al., 2016). Conversely, protein sequences naturally fold into higher-order three-dimensional (3D) structures, and distinct structures typically confer distinct functions (Gao et al., 2020). Mapping between sequence and structure in both directions is therefore a central theme in protein science (Kuhlman & Baker, 2000). The forward direction is the well-known protein folding task, which predicts structure (and hence function) from sequence (Jumper et al., 2021; Baek et al., 2021; Abramson et al., 2024). In this work, we focus on the opposite direction, *protein inverse folding*, which aims to design an amino acid sequence that will stably fold into a specified 3D structure (Dauparas et al., 2022; Gao et al., 2022a; Yi et al., 2023; Ren et al., 2024), enabling applications such as enzyme engineering and drug discovery. For example, to add a new binding site on the surface of an existing enzyme, researchers first design the local 3D structure required for binding, then use inverse folding methods to derive candidate sequences for synthesis and experimental validation.

Traditional protein inverse folding methods directly encode the 3D structure and perform position-wise amino acid (residue) classification (one-to-one mapping), as shown in Figure 1(a). Constrained by dataset scale and model capacity, their performance is limited. Besides, since the structure encoder ignores sequence context, the resulting sequences are usually biologically unreasonable. With the advent of large pretrained protein sequence models, i.e., protein language models (PLMs) (Rives et al., 2021; Lin et al., 2023; Madani et al., 2023; Elnaggar et al., 2021; Brandes et al., 2022), recent methods (Gao et al., 2023; Zhu et al., 2024) often append a post-hoc refinement stage that uses PLMs to revise the output of the structure encoder. For example, as shown in Figure 1(b), if the encoder proposes "...LISE..." but assigns low confidence to "I", the PLM could leverage pretrained sequence knowledge to decide whether "I" should be replaced by another residue, conditioning on the context of "I", thereby improving sequence plausibility. However, such refinement is decoupled from the original structural evidence and is inherently upper-bounded by the first-stage output: when the proposed sequence already violates structural constraints, PLM edits may further degrade structural compatibility, compounding errors for inverse folding.

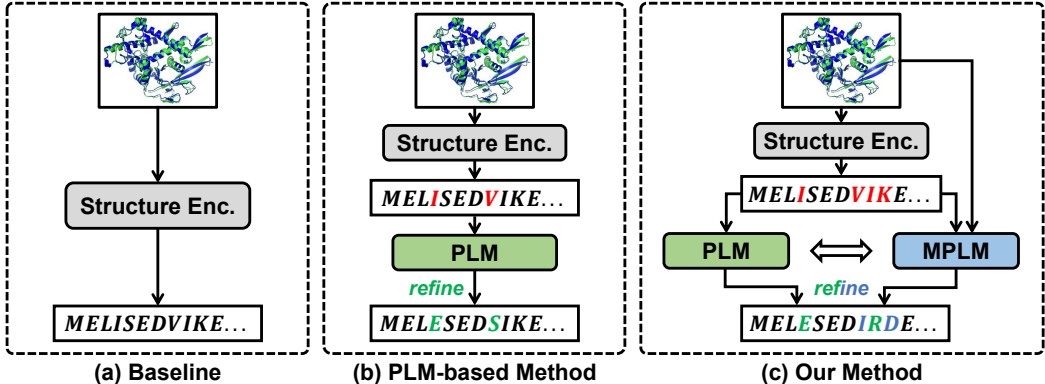

Figure 1: Comparison of various inverse folding frameworks: (a) The baseline model is trained from scratch without utilizing any pretrained knowledge to directly predict amino acid sequence from the protein structure. (b) PLM-based methods refine the output sequence of the structure encoder with pretrained sequence knowledge. (c) Our proposed DualFold symmetrically utilizes the pretrained knowledge of sequence and structure to refine the coarse sequence.

To bridge the gap between pretrained knowledge and given structures during refinement, we find that recent large pretrained multimodal protein models, i.e., multimodal protein language models (MPLMs) (Hayes et al., 2025; Wang et al., 2024; Hsieh et al., 2025), can directly map 3D structures to sequences and are naturally suited to the inverse folding task, even in a zero-shot manner. However, in practice, we find that MPLMs used in isolation are ineffective for this task, perhaps due to structural distribution shifts between pretraining and this task or their limited sequence modeling capacity. Therefore, we introduce **DualFold**, a harmonic dual-path framework that fuses PLMs and MPLMs in parallel to provide complementary pretrained knowledge for sequence generation. Specifically, as shown in Figure 1(c), our method first follows the traditional approach to derive a coarse sequence via a structure encoder, then we employ PLMs to provide pretrained sequence context knowledge, and meanwhile, leverage MPLMs to provide pretrained structural knowledge to refine the proposed sequence. Unlike the traditional uses of PLMs, our balanced dual-path refinement simultaneously accounts for contextual dependencies and the residue-specific positional information, thereby addressing the refinement gap.

Furthermore, because the dependence on structure and sequence context varies across residues, we design an adaptive per-residue fusion module that dynamically integrates structure cues, context dependencies, and structural priors so that the refinement can maintain compatibility with residue-specific biochemical characteristics. We also introduce a self-correction iterative training strategy to align training with the inference routine. A structure encoder first proposes a coarse sequence, PLMs and MPLMs perform an initial refinement, and then stochastically mix the outputs with the coarse sequence before the final refinement stage. This iterative procedure mirrors the test-time iterative prediction manner and reduces the mismatch between training and testing.

On standard inverse folding benchmarks, including the widely used CATH (Orengo et al., 1997), TS50, and TS500 (Li et al., 2014), our method sets new state-of-the-art performance. Extensive ablations corroborate our findings and validate the proposed design, revealing the limitations of stand-alone MPLMs, the benefit of our dual-path architecture, and the effectiveness of our adaptive integration and self-correction modules. Our main contributions are summarized as follows:

- We diagnose the shortcomings of post-hoc refinement in current methods and argue that effective refinement must incorporate raw 3D evidence.

- We introduce MPLMs into the inverse folding and empirically show that MPLMs alone are insufficient for this task.

- We propose DualFold, which combines pretrained sequence and structural knowledge via adaptive fusion and self-correction for sequence generation, achieving SOTA performance across multiple benchmarks and informing future directions.

## 2 RELATED WORK

**Protein Inverse Folding**. Early approaches in protein inverse folding primarily relied on physics-based methods such as Rosetta Design (Liu & Kuhlman, 2006), which searched for low-energy sequences compatible with target backbones via sampling strategies like Monte Carlo simulation (Kuhlman & Bradley, 2019). While foundational, these methods were limited by approximate energy functions and high computational costs. The advent of deep learning (Kingma & Welling, 2013; Goodfellow et al., 2014; Vaswani et al., 2017; LeCun et al., 2002; Devlin et al., 2019; Sohl-Dickstein et al., 2015) enabled the use of geometric deep learning, where protein structures are modeled as graphs and processed with GNNs or equivariant networks to map structures to sequences. Representative works include GraphTrans (Ingraham et al., 2019), GVP (Jing et al., 2020), and ProteinMPNN (Dauparas et al., 2022), which achieved strong sequence recovery through optimized graph representations and message passing. PiFold (Gao et al., 2022a) further improved efficiency with non-autoregressive predictions. Despite these advances, purely geometric models rely on limited structure-sequence pairs and struggle to capture evolutionary covariation present in massive sequence databases. To address this, recent methods couple protein language models (PLMs) such as the ESM series (Rives et al., 2021; Lin et al., 2023) with geometric models. Typically, a geometric model first generates sequences, followed by refinement with PLMs, as in LM-Design (Zheng et al., 2023), Knowledge-Design (Gao et al., 2023), and Bridge-IF (Zhu et al., 2024). While effective, this serial pipeline suffers from decoupled refinement where PLMs lack access to structural constraints. In contrast, our work introduces a parallel dual-path refinement, ensuring refined sequences remain faithful to both evolutionary patterns and target structures.

**Multimodal Protein Language Models (MPLMs)**. Recent MPLMs aim to unify sequences and structures under a single framework. ESM-3 (Hayes et al., 2025) embodies this direction by jointly modeling sequences, atomic coordinates, and functional annotations with Transformers, though its discretization of continuous coordinates constrains its direct application to inverse folding. Other notable MPLMs include DPLM-2 (Wang et al., 2024), a diffusion-based framework capable of handling discrete and structural modalities, and Evola (Zhou et al., 2025), an 80B-parameter model that integrates protein-specific knowledge for design and functional tasks. These models highlight the promise of MPLMs for generative protein tasks, though challenges remain in fully exploiting them for inverse folding. In this study, we harness the structure-aware capabilities of MPLMs through our proposed framework, further unlocking their potential for such tasks.

## 3 METHOD

### 3.1 PRELIMINARIES

**Baseline**. Protein inverse folding seeks to recover an amino acid sequence $\boldsymbol{s} = (s_1, s_2, \ldots, s_\ell)$ from a given protein structure $\boldsymbol{x} = (x_1, x_2, \ldots, x_\ell)$, where $s_i \in \mathcal{S}$ denotes the residue identity at position $i$, $\mathcal{S}$ is the set of amino acids, and $x_i$ denotes the atomic coordinates for residue $i$. The one-hot encoding of $s_i$ is $\boldsymbol{y}_i \in \{0, 1\}^{|\mathcal{S}|}$ with components $(\boldsymbol{y}_i)_a = \mathbb{I}\{a = s_i\}, a \in \mathcal{S}$, where $\mathbb{I}$ is the indicator function. Given training set $\mathcal{D} = \{(\boldsymbol{x}, \boldsymbol{y})\}$, the baseline methods learn a structure encoder $f_e$ producing per-position prediction (with entire structure of the protein), and the per-sample training objective could be written as sequence-averaged cross-entropy loss:

$$\mathcal{L}_{\text{base}} = -\frac{1}{\ell} \sum_{i=1}^{\ell} \boldsymbol{y}_i \cdot \log \frac{\exp(f_e(\boldsymbol{x})_i)}{\sum_k \exp(f_e(\boldsymbol{x})_{i,k})}, \tag{1}$$

where $\exp(\cdot)$ is the exponential, $f_e(\boldsymbol{x})_i \in \mathbb{R}^{|\mathcal{S}|}$ is the predicted logits for $s_i$, and $k$ indexes classes of $\mathcal{S}$. In training, one minimizes the empirical mean of $\mathcal{L}_{\text{base}}$ over all samples in the dataset $\mathcal{D}$.

**PLM-based Method**. To incorporate sequence evolutionary knowledge, recent methods use a protein language model (PLM) $f_s$ to refine the predictions of an optimized structure encoder $f_e^*$. Specifically, they first train a structure encoder $f_e$ via Eq. (1), and then finetune $f_s$ conditioning on the outputs of $f_e^*$, to conduct post hoc refinement. The sequence-level training objective is:

$$\mathcal{L}_{\text{plm}} = -\frac{1}{\ell} \sum_{i=1}^{\ell} \boldsymbol{y}_i \cdot \log \frac{\exp(f_s(\hat{\boldsymbol{s}})_i)}{\sum_k \exp(f_s(\hat{\boldsymbol{s}})_{i,k})}, \quad \hat{\boldsymbol{s}}_j = \operatorname*{argmax}_k f_e^*(\boldsymbol{x})_{j,k}, \tag{2}$$

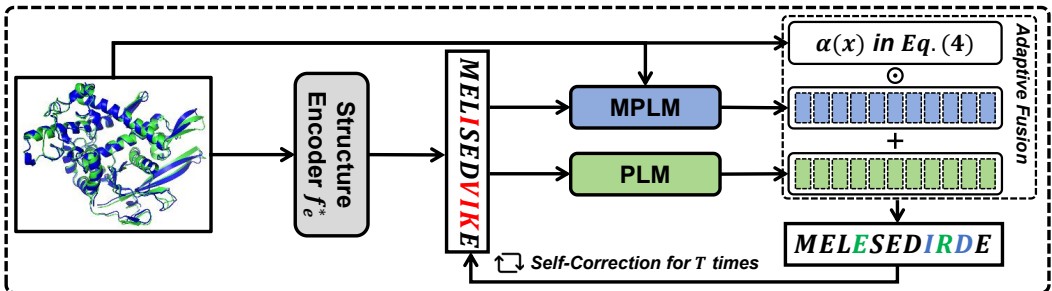

Figure 2: **Overview of DualFold**. DualFold employs a dual-path refinement framework integrating structural priors from MPLMs and evolutionary priors from PLMs. Starting with a pre-trained structure encoder $f_e^*$, initial predictions are refined through parallel structure and sequence branches. The adaptive fusion module combines these signals using structure-aware coefficients $\alpha(\boldsymbol{x})$ in Eq.( 4), $\odot$ denotes element-wise product, followed by iterative self-correction to align training with inference.

where $\hat{\boldsymbol{s}}$ is the output of the optimal (and frozen) structure encoder $f_e^*$, the argmax is taken over the class index $k$ for each residue $j$, and $f_s$ is trainable during finetuning. Note that, since no structure information $\boldsymbol{x}$ is available to $f_s$ during refinement, the post-hoc editing is inherently limited by $f_e^*$.

## 3.2 OUR METHOD — DUALFOLD

**Overview**. To address the limitations of structure-ignorant refinement in PLM-based methods, we introduce DualFold, a symmetric dual-path architecture that couples structural priors from multimodal protein language models (MPLMs) with evolutionary priors from PLMs for stronger refinement. As shown in Figure 2, we first follow standard practice to train a structure encoder $f_e$ by Eq. (1). We then encode structure knowledge via an MPLM and sequence knowledge via a PLM, and use them to adjust the sequence predictions through an *adaptive fusion* module. A *self-correction* scheme is further proposed to align training and inference to mitigate exposure bias and reduce error accumulation. Together, these components enable a more harmonious and effective use of pretrained knowledge for protein inverse folding.

**Dual-Path Refinement**. Following PLM-based practice, we first train a structure encoder via Eq. (1) to obtain $f_e^*$. We then incorporate *sequence* priors using a PLM $f_s$ based on the predicted sequence $\hat{\boldsymbol{s}} = \mathrm{argmax}\, f_e^*(\boldsymbol{x})$ and incorporate *structure* priors using an MPLM $f_m$ conditioned on $\boldsymbol{x}$. These two signals are fused by a symmetric dual-path module $f_{\text{dual}}$, and the per-sample training objective of our dual-path refinement is:

$$\mathcal{L}_{\text{ours}} = -\frac{1}{\ell} \sum_{i=1}^{\ell} \boldsymbol{y}_i \cdot \log \frac{\exp(f_{dual}(f_s, \hat{\boldsymbol{s}}, f_m, \boldsymbol{x})_i)}{\sum_k \exp(f_{dual}(f_s, \hat{\boldsymbol{s}}, f_m, \boldsymbol{x})_{i,k})}, \tag{3}$$

where $\exp(\cdot)$ is the exponential function, $\cdot$ is the dot product, $k$ indexes classes of $\mathcal{S}$. To effectively fuse pretrained knowledge from both paths, we introduce an Adaptive Fusion Module that dynamically weights and blends their complementary cues. In addition, because inference aggregates more information sources, we propose a Self-Correction strategy that incrementally incorporates the dual paths step by step, thereby mitigating the train–test discrepancy.

**Adaptive Fusion**. Unlike PLM-based refinement, which derives priors solely from sequence context, our dual-path refinement jointly leverages sequence context and structure signals. Since some residues are tightly constrained by local geometry while others depend more on long-range sequence dependencies, we design a residue-wise Adaptive Fusion module that dynamically integrates these complementary cues:

$$f_{dual}(f_s, \hat{\boldsymbol{s}}, f_m, \boldsymbol{x})_i = f_s(\hat{\boldsymbol{s}})_i + \alpha(\boldsymbol{x})_i \cdot f_m(\boldsymbol{x}, \hat{\boldsymbol{s}})_i, \tag{4}$$

where $\hat{\boldsymbol{s}}$ denotes the sequence predicted by the frozen optimal structure encoder $f_e^*$, $\cdot$ is the dot product, and $f_s$ and $f_m$ could be fine-tuned during training. The residue-wise structure-aware coefficient $\alpha(\boldsymbol{x})$ controls how strongly residue $i$ is influenced by the corresponding structure and could be realized as $\alpha(\boldsymbol{x}) = \boldsymbol{w}^\top f_e^*(\boldsymbol{x})$, where $\boldsymbol{w}$ is a trainable projection vector.

**Self-Correction**. Current PLM-based methods typically perform iterative generation at inference, feeding the predicted sequence back into the PLM for several rounds, which induces a mismatch between one-step training and multi-step inference. Under our dual-path framework, this discrepancy can be amplified since multiple knowledge sources are fused at each step. To address this, we introduce a Self-Correction strategy that unrolls the dual-path refinement during training to mirror the iterative inference procedure. The iterative process could be written as:

$$\hat{\boldsymbol{s}}^{(0)} = \operatorname*{argmax} f_e^*(\boldsymbol{x}) \tag{5}$$

$$\hat{\boldsymbol{s}}_i^{(t)} = \operatorname*{argmax}_k f_{dual}(f_s, \hat{\boldsymbol{s}}^{(t-1)}, f_m, \boldsymbol{x})_{i,k}, \tag{6}$$

where $t = 1, \ldots, T$ is iterative step and $T$ is the total step. During training, we compute the loss using the refined prediction at $T$ for the loss calculation.

## 4 EXPERIMENTS

### 4.1 EXPERIMENTAL PROTOCOL

**Datasets.** We train our model on CATH4.2 and CATH4.3. The CATH4.2 dataset consists of 18,024 proteins for training, 608 proteins for validation, and 1,120 proteins for testing, following the same data splitting as GraphTrans (Ingraham et al., 2019). The CATH4.3 dataset includes 16,153 structures for training, 1,457 for validation, and 1,797 for testing, following the same data splitting as ESMIF (Hsu et al., 2022).

For comprehensive evaluation, we assess our model on multiple protein structure datasets. We test on the CATH4.2 test set with 1,120 proteins and the CATH4.3 test set with 1,797 proteins to measure performance on protein folds similar to those seen during training. Additionally, we evaluate on the TS50 and TS500 datasets, comprising 50 and 500 proteins respectively as established by (Li et al., 2014). To examine our model's generalization capabilities, we further test on the challenging CASP15 (Elofsson, 2023) and CASP16 (Yuan et al., 2025) monomeric tertiary structure targets, which include 45 and 50 proteins respectively and represent novel protein structures. The specific protein identifiers and their official release times for CASP15 and CASP16 are provided in Appendix A.3 to facilitate reproducibility and cross-study comparison.

**Baseline Models and Evaluation Metrics.** We compare DualFold with recent graph-based models (StructGNN, GraphTrans (Ingraham et al., 2019), GCA (Tan et al., 2022), GVP (Jing et al., 2020), GVP-large (Hsu et al., 2022), AlphaDesign (Gao et al., 2022b), ESM-IF (Hsu et al., 2022), ProteinMPNN (Dauparas et al., 2022),PiFold (Gao et al., 2022a),GraDe-IF (Yi et al., 2023)), and PLM-based optimized models (LM-Design (Zheng et al., 2023), Knowledge-Design (Gao et al., 2023), Bridge-IF (Zhu et al., 2024)). We report perplexity and median recovery rate to assess performance(computation details in Appendix A.6), with evaluations on the CATH dataset divided into three protein types: Short proteins (length $\leq$ 100), Single-chain proteins (with only 1 chain in CATH), and All proteins.

**Implementation Details.** DualFold employs a frozen pre-trained PiFold encoder for fundamental structural representation, while only the adaptive fusion MLP is fully fine-tuned. As language modeling backbones, we use the open-source ESM-3 (1.4B, MPLM) and its co-trained counterpart ESM-C (600M, PLM) (Hayes et al., 2025). LoRA (Hu et al., 2022) is applied solely to MPLM and PLM with rank $r = 8$, scaling factor $\alpha = 32$, and dropout 0.1, leading to ~0.1% trainable parameters in total (further configuration details are provided in the Appendix A.5). Training is performed on a single NVIDIA A800 GPU with batch size 4, cosine learning rate scheduling, and typically converges within 5 epochs. For the self-correction training strategy, we set $T = 2$ (more details about the inference phase setting of $T$ are in Appendix A.2). All results are reported based on this final configuration.

### 4.2 RESULTS AND ANALYSIS

Through the following Q&A, we provide in-depth discussions of the experimental results for protein inverse folding, offering a comprehensive analysis of our DualFold model's performance across various benchmarks. We address key questions regarding performance comparisons, architectural

Table 1: Results comparison on the CATH4.2 and CATH4.3 datasets. Some benchmarked results are quoted from (Gao et al., 2023; Zhu et al., 2024). Perplexity (↓) indicates sequence prediction uncertainty, where lower values are better; Recovery (%↑) measures sequence accuracy, where higher is better. The best and suboptimal results are labeled with bold and underline.

| | Model | Perplexity ↓ | | | Recovery % ↑ | | |
|---|---|---|---|---|---|---|---|
| | | Short | Single-chain | All | Short | Single-chain | All |
| **CATH4.2** | StructGNN | 8.29 | 8.74 | 6.40 | 29.44 | 28.26 | 35.91 |
| | GraphTrans | 8.39 | 8.83 | 6.63 | 28.14 | 28.46 | 35.82 |
| | GCA | 7.09 | 7.49 | 6.05 | 32.62 | 31.10 | 37.64 |
| | GVP | 7.23 | 7.84 | 5.36 | 30.60 | 28.95 | 39.47 |
| | AlphaDesign | 7.32 | 7.63 | 6.30 | 34.16 | 32.66 | 41.31 |
| | ProteinMPNN | 6.21 | 6.68 | 4.61 | 36.35 | 34.43 | 45.96 |
| | PiFold | 6.04 | 6.31 | 4.55 | 39.84 | 38.53 | 51.66 |
| | GraDe-IF | 5.49 | 6.21 | 4.35 | 45.27 | 42.77 | 52.21 |
| | LM-Design | 5.66 | 5.52 | 4.01 | 46.84 | 48.63 | 56.63 |
| | Knowledge-Design | 5.48 | 5.16 | 3.46 | 44.66 | 45.45 | 60.77 |
| | Bridge-IF | 5.68 | 5.06 | 3.83 | 43.86 | 48.96 | 58.59 |
| | **DualFold** | **4.69** | **4.01** | **3.23** | **50.00** | **55.45** | **63.11** |
| **CATH4.3** | GVP-large | 7.68 | 6.12 | 6.17 | 32.60 | 39.40 | 39.20 |
| | ESM-IF | 8.18 | 6.33 | 6.44 | 31.30 | 38.50 | 38.30 |
| | +1.2M AF2 predicted data | 6.05 | 4.00 | 4.01 | 38.10 | 51.50 | 51.60 |
| | ProteinMPNN | 6.35 | 6.25 | 4.89 | 40.73 | 41.18 | 47.69 |
| | PiFold | 5.50 | 5.76 | 4.44 | 43.84 | 44.32 | 50.62 |
| | LM-Design | 5.66 | 5.52 | 4.01 | 42.84 | 43.69 | 55.68 |
| | Knowledge-Design | 5.47 | 5.23 | 3.49 | 43.89 | 45.95 | 60.38 |
| | Bridge-IF | 5.17 | 4.63 | 3.68 | 50.00 | 53.49 | 58.93 |
| | **DualFold** | **4.26** | **3.91** | **3.22** | **55.81** | **58.20** | **62.23** |

innovations, generalization capabilities, ablation studies, and biological plausibility to systematically evaluate our approach's contributions to the field.

**Q1. How does DualFold perform on multiple benchmarks?**

As shown in Table 1, DualFold achieves state-of-the-art performance across all evaluated benchmarks. On CATH4.2, it reaches a 63.11% overall sequence recovery rate, outperforming the best previous PLM-based method, Knowledge-Design, by 2.34% and exceeding structure-only methods such as PiFold by 11.45 percentage points. The gains are especially notable for diverse protein categories: recovery on short proteins (length $\leq$ 100) reaches 50.00%, an improvement of 5.34% over Knowledge-Design, while performance on single-chain proteins rises to 55.45%, 6.49% higher than Bridge-IF. These results highlight DualFold's effectiveness in leveraging both sequence and structural information, particularly for structurally constrained proteins. On CATH4.3, the model maintains its advantage with a recovery rate of 62.23%, showing consistent improvements across datasets. Furthermore, DualFold achieves the lowest perplexity across all protein categories, indicating more confident and accurate sequence predictions.

As shown in Table 2, DualFold establishes new state-of-the-art results on both TS50 and TS500, which are more diverse than CATH. On TS50, it achieves a perplexity of 2.84 and recovery of 66.02%, surpassing Knowledge-Design by 3.23 percentage points in recovery and reducing perplexity by 0.26. On the larger TS500, DualFold reaches 70.48% recovery with the lowest perplexity of 2.74. These consistent improvements, with perplexity reductions of 8.4% (TS50) and 4.2% (TS500), demonstrate both scalability and robustness of the dual-stream architecture in capturing sequence–structure relationships.

To further validate the generalization and generative capabilities of our model, we evaluated DualFold on the CASP15 and CASP16 datasets, which are drawn from the Critical Assessment of protein Structure Prediction (CASP) (Yuan et al., 2025) competitions and provide a standardized,

Table 2: Results comparison on the TS50&TS500 dataset. Some benchmarked results are quoted from (Gao et al., 2023). The best and suboptimal results are labeled with bold and underline.

| Model | TS50 | | TS500 | |
|---|---|---|---|---|
| | Perplexity ↓ | Recovery% ↑ | Perplexity ↓ | Recovery% ↑ |
| StructGNN | 5.40 | 43.89 | 4.98 | 45.69 |
| GraphTrans | 5.60 | 42.20 | 5.16 | 44.66 |
| GVP | 4.71 | 44.14 | 4.20 | 49.14 |
| GCA | 5.09 | 47.02 | 4.72 | 47.74 |
| AlphaDesign | 5.25 | 48.36 | 4.93 | 49.23 |
| ProteinMPNN | 3.93 | 54.43 | 3.53 | 58.08 |
| PiFold | 3.86 | 58.72 | 3.44 | 60.42 |
| LM-Design | 3.50 | 57.89 | 3.19 | 63.65 |
| Knowledge-Design | 3.10 | 62.79 | 2.86 | 69.19 |
| **DualFold** | **2.84** | **66.02** | **2.74** | **70.48** |

Table 3: Comparison of results on CASP15 and CASP16 datasets. The best and suboptimal results are labeled with bold and underline.

| Model | CASP15 | | CASP16 | |
|---|---|---|---|---|
| | Perplexity ↓ | Recovery% ↑ | Perplexity ↓ | Recovery% ↑ |
| ProteinMPNN | 5.69 | 43.06 | 7.19 | 39.32 |
| PiFold | 4.87 | 48.45 | 5.66 | 47.10 |
| LM-Design | 5.12 | 50.28 | 5.79 | 48.64 |
| **DualFold** | **3.90** | **56.57** | **4.50** | **52.18** |

highly challenging benchmark for protein structure modeling, with CASP16 representing the latest experimentally determined protein structures. As shown in Table 3, DualFold demonstrates consistent and substantial improvements over previous state-of-the-art methods across both benchmarks. On CASP15, our model achieves 56.57% recovery rate, while on CASP16, it reaches 52.18%, representing improvements on both datasets. These results demonstrate DualFold's exceptional predictive accuracy and robust generalization capabilities when handling the most challenging novel protein structures.

**Q2. Are the various module designs in DualFold reasonable and valuable?**

**Ablations for our dual-path design**. We conduct ablation studies to validate our dual-stream design by evaluating four configurations: (1) the complete DualFold with both streams, (2) DualFold without the PLM branch (w/o PLM), and (3) DualFold without the MPLM branch (w/o MPLM).

The results in Table 4 demonstrate the synergistic benefits of our dual-stream architecture. The full DualFold model achieves the best performance across all metrics, with perplexity of 3.23 and recovery rate of 63.11% on the full test set. When we ablate the PLM branch (w/o PLM), keeping only the MPLM stream, performance drops significantly. Similarly, ablating the MPLM branch (w/o MPLM) while keeping only the PLM stream also degrades performance. These results confirm that both structural and sequence information are essential and complementary.

Notably, when we directly evaluate a pretrained MPLM in a zero-shot setting on CATH4.2—without any task-specific adaptation—it only achieves a perplexity of 6.50 and a recovery rate of 42.03% on the full test set. This raw performance is substantially worse than both single-stream ablations, emphasizing that although MPLMs encode rich structural priors, they cannot be directly applied to inverse folding. Effective adaptation is essential to unlock their potential. By contrast, our dual-stream design with joint training leverages the complementary strengths of PLMs and MPLMs, yielding significantly stronger results. For completeness, Table 7 in the Appendix A.1 reports additional comparisons with integrating MPLM under another mainstream architecture.

Table 4: Ablation results for the dual-stream architecture. The full DualFold outperforms both single-stream variants, confirming the complementary roles of PLM and MPLM.

| Model Variant | Perplexity ↓ | | | Recovery % ↑ | | |
|---|---|---|---|---|---|---|
| | Short | Single-chain | All | Short | Single-chain | All |
| DualFold | **4.69** | **4.01** | **3.23** | **50.00** | **55.45** | **63.11** |
| *w/o* PLM | 6.11 | 5.66 | 4.05 | 42.55 | 45.21 | 56.62 |
| *w/o* MPLM | 5.51 | 4.73 | 3.61 | 44.02 | 50.89 | 59.56 |

Table 5: Ablation study of the three proposed components in DualFold: Prior (frozen pre-trained structure encoder), AF (Adaptive Fusion of expert predictions), and SC (Self-Correction via iterative refinement). ✓ indicates the component is enabled, and × indicates it is disabled.

| Prior | AF | SC | Perplexity ↓ | | | Recovery % ↑ | | |
|---|---|---|---|---|---|---|---|---|
| | | | Short | Single-chain | All | Short | Single-chain | All |
| × | × | × | 4.98 | 4.40 | 3.46 | 49.52 | 53.00 | 61.26 |
| × | ✓ | ✓ | 4.92 | 4.32 | 3.40 | 48.56 | 53.26 | 61.85 |
| ✓ | × | ✓ | 4.83 | 4.20 | 3.35 | 49.75 | 54.52 | 62.31 |
| ✓ | ✓ | × | 4.80 | 4.18 | 3.32 | 49.81 | 54.60 | 62.43 |
| ✓ | ✓ | ✓ | **4.69** | **4.01** | **3.23** | **50.00** | **55.45** | **63.11** |

**Ablations for our proposed modules**. Table 5 presents an ablation study of the three key components in DualFold: the Prior training strategy, Adaptive Fusion (AF), and Self-Correction (SC). The Prior strategy employs a frozen pre-trained structure encoder to ensure stable and informative feature extraction. Adaptive Fusion dynamically integrates predictions from multiple expert modules during sequence decoding, optimizing the generation process by selectively enhancing the most reliable signals. Self-Correction utilizes an iterative refinement mechanism to progressively correct errors in intermediate predictions, thereby mitigating error propagation and improving the final output quality.

The results clearly demonstrate the contribution of each component. Introducing the Prior strategy consistently lowers perplexity and enhances recovery by providing stable structural representations. Adaptive Fusion further refines predictions by leveraging complementary signals from different expert modules, while Self-Correction progressively mitigates error propagation through iterative refinement. When combined, these components work in concert to deliver the lowest perplexity and highest recovery across all categories, underscoring their complementary roles in strengthening both the accuracy and robustness of protein sequence generation.

### Q3. Is DualFold model-agnostic?

We evaluated the DualFold framework's compatibility with different foundation models by integrating two distinct structure encoders: ProteinMPNN and PiFold. These experiments were conducted under controlled conditions, utilizing the same MPLM and PLM components while maintaining consistent training strategies. As shown in Table 6, our framework consistently enhances the performance of both encoders across all evaluated categories. DualFold achieves substantial improvements in sequence recovery for both ProteinMPNN (from 49.87% to 61.45%) and PiFold (from 51.66% to 63.11%), with corresponding perplexity reductions. Notably, when equipped with the more advanced PiFold encoder, DualFold achieves superior performance across all metrics, indicating that better structure encoders provide more precise prior conditions for the synergistic generation process. These results confirm that the DualFold architecture is fundamentally model-agnostic while scaling its performance with the quality of underlying components.

### Q4. Are the generated sequences biologically plausible?

To evaluate the biological plausibility of sequences generated by DualFold, we conducted a validation experiment using AlphaFold3 (Abramson et al., 2024) for structure prediction. We selected two proteins from CASP16 with varying lengths: T1234 (377 residues) and T1266 (295 residues).

Table 6: Performance comparison of DualFold when using ProteinMPNN versus PiFold as the structure encoder on CATH4.2.

| Structure Encoder | Perplexity ↓ | | | Recovery % ↑ | | |
|---|---|---|---|---|---|---|
| | Short | Single-chain | All | Short | Single-chain | All |
| ProteinMPNN | 6.21 | 6.68 | 4.57 | 36.35 | 34.43 | 49.87 |
| *with* DualFold | 5.02 | 4.56 | 3.34 | 48.96 | 52.28 | 61.45 |
| PiFold | 6.04 | 6.31 | 4.18 | 39.84 | 38.53 | 51.66 |
| *with* DualFold | 4.69 | 4.01 | 3.23 | 50.00 | 55.45 | 63.11 |

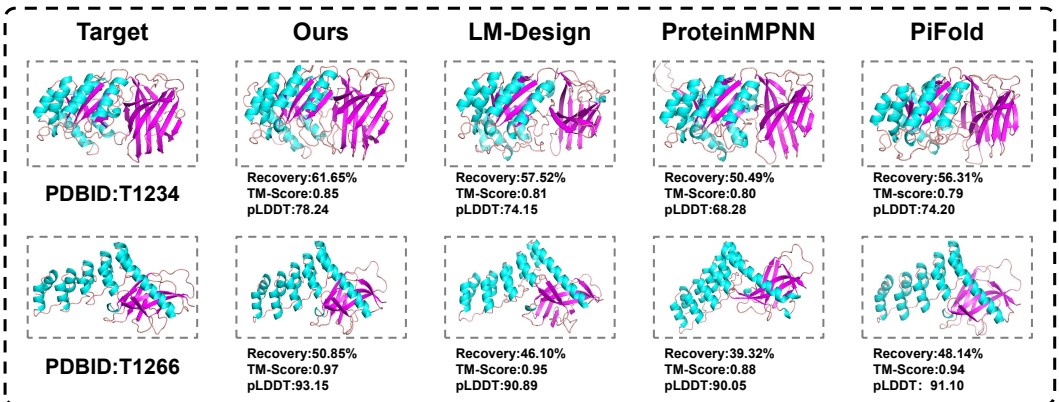

Figure 3: We compare sequence recovery, structure prediction confidence (pLDDT), and structural similarity (TM-score) (Zhang & Skolnick, 2005) for sequences generated by DualFold, LM-Design, ProteinMPNN and PiFold. Note that TM-score ranges from 0 to 1 (higher is better), and pLDDT ranges from 0 to 100 (higher is better).

For each target structure, we generated sequences using DualFold, LM-Design, PiFold, and Protein-MPNN, then used AlphaFold3 to predict structures from these designed sequences.

As shown in Figure 3, DualFold-generated sequences exhibit stronger consistency with the intended target structures across recovery, pLDDT, and TM-score. For T1234, our method achieves a sequence recovery of 61.65%, higher than LM-Design (57.52%), ProteinMPNN (50.49%), and PiFold (56.31%). For T1266, DualFold attains 50.85% compared to 46.10%, 39.32%, and 48.14% for the respective baselines. AlphaFold3 predictions suggest that sequences designed by DualFold yield more confident folding models (pLDDT: 78.24 for T1234, 93.15 for T1266) with better structural alignment to the native backbone (TM-scores of 0.85 and 0.97, respectively).

Additional visual comparisons on four more CASP16 targets (T1214, T1235, T1299, T1259) are provided in Appendix A.4. These results provide preliminary in silico evidence that our dual-stream architecture can improve not only recovery metrics but also the predicted structural plausibility of designed sequences. While AlphaFold-derived scores cannot substitute for experimental verification, they offer encouraging indications that DualFold-generated sequences are more compatible with the desired folds.

## 5 CONCLUSION & LIMITATION

We introduce DualFold, a novel symmetric dual-stream framework for protein inverse folding that addresses the limitations of serial architectures by fostering continuous interaction between structural and evolutionary knowledge. Our adaptive fusion and self-correction mechanisms enhance sequence design, achieving state-of-the-art performance across benchmarks. Despite these advances, challenges remain. Our evaluations primarily rely on in silico metrics, and the limited extent of wet-lab validation leaves the functional viability of the generated sequences unverified. Future work will explore stronger foundation models to push the boundaries of computational protein design.

# 6 STATEMENT

**Ethics Statement**. This research presents a symmetric dual-path integration framework for protein inverse folding using pretrained protein language models. The work is entirely computational, involving no human subjects, animal testing, or sensitive data. While the method may support future advances in biotechnology and medicine, we identify no foreseeable risks of misuse, bias, or security concerns, as it is limited to generating amino acid sequences consistent with given structures. The study received no external sponsorship or conflicts of interest, and all authors followed the ICLR Code of Ethics.

**Reproducibility Statement**. Our implementation of the proposed DualFold architecture is fully reproducible. All required modules and models are publicly available, and all evaluation metrics (perplexity and recovery rate) are computed following standard protocols. The datasets used are publicly accessible, with details of specific dataset splits provided in the experimental section and appendix. The complete code will be released on GitHub upon acceptance of the paper.

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

# A APPENDIX

## A.1 COMPARATIVE ARCHITECTURE DESIGNS WITH MPLM

To further investigate the role of cross-modal integration of MPLMs in protein inverse folding, we replaced the original protein language model (PLM) in the LM-Design framework with an MPLM. By systematically evaluating two representative adapter schemes within this framework, this study thoroughly examines how different connection strategies between multimodal protein language models and structure encoders affect downstream recovery performance, thereby providing clearer insights into the mechanisms of synergy across modalities.We evaluated two representat:

(1) Structure-Aware Adapter (original LM-Design implementation): This variant integrates high-dimensional embeddings from a protein structure encoder through lightweight adapter modules.

Cross-modal interaction is realized via multi-head attention, explicitly mixing coordinate-based information with sequence embeddings.

(2) Parameter-Tuning Adapter (modified version): Here, the attention-based fusion is removed. The adapter reduces to simple trainable MLP layers, leaving structural sensitivity to be modeled implicitly by the MPLM itself without direct feature injection.

Both architectures were trained under identical setups (datasets, losses, optimizers, and hyperparameters). We further controlled two experimental factors: (i) *Training Strategy* – training the structure encoder from scratch versus freezing pretrained encoder parameters; (ii) *Input Modality* – full multimodal input (amino-acid sequence + 3D coordinates) versus sequence-only input.

Table 7: Recovery performance comparison of adapter architectures with MPLM backbone under different training strategies and input modalities.

| Training Strategy | Architecture | Full Input | Sequence Only |
|---|---|---|---|
| From Scratch | Struct-Aware Adapter | 52.46 | 53.68 |
| | Param-Tuning Adapter | **53.60** | 53.12 |
| Pretrained + Frozen | Struct-Aware Adapter | 53.95 | **54.56** |
| | Param-Tuning Adapter | **55.33** | 54.01 |

The results reveal a non-trivial pattern. Despite its richer multimodal design, the Structure-Aware Adapter does not consistently outperform the simpler Parameter-Tuning variant. In fact, under full-input conditions (sequence + coordinates), the more "sophisticated" attention fusion can underperform, while the streamlined variant often achieves stronger recovery rates. This counterintuitive outcome illustrates a phenomenon we term the *modality paradox*: providing richer multimodal inputs does not automatically translate into better performance, and may even degrade results due to feature redundancy, optimization conflicts, or discrepancies in how modalities are aligned. In contrast, the Parameter-Tuning Adapter leverages MPLM's structurally sensitive embeddings more stably, demonstrating that effective adaptation requires *carefully crafted integration mechanisms*. Overall, these findings underscore an important insight for the field: when fusing pretrained MPLMs for inverse folding, more information is not always better—thoughtful architecture design is essential.

## A.2 ITERATIVE INFERENCE ANALYSIS

We analyzed the impact of iterative refinement steps ($T$) on DualFold's performance with and without Self-Correction. Table 8 shows the results on CATH4.2.

Table 8: Impact of iterative refinement steps ($T$) on sequence recovery rate (%).

| $T$ | Recovery(%) ↑ | |
|---|---|---|
| | With Self-Correction | Without Self-Correction |
| 1 | 62.25 | 60.57 |
| 2 | **63.11** | 61.19 |
| 3 | 62.96 | 62.23 |
| 4 | 62.96 | **62.43** |
| 5 | 63.10 | 62.42 |

The results clearly demonstrate the effectiveness of the Self-Correction mechanism. With Self-Correction, the model achieves superior performance across all iteration counts, with the most significant improvements at lower iterations. Notably, Self-Correction enables the model to reach peak performance (63.11%) early in the refinement process. In contrast, without Self-Correction, the model shows slower convergence and lower overall performance, never reaching the accuracy achieved by Self-Correction even with more iterations.

## A.3 CASP15 AND CASP16 PROTEIN DETAILS

For the purpose of ensuring reproducibility, this appendix summarizes the specific protein targets used in our experiments from the CASP15 and CASP16 datasets. Tables 9 and 10 provide the official release names and dates of each protein. This collection allows future studies to easily identify and cross-reference proteins with their corresponding release times.

Table 9: CASP15 protein names and release times.

| Protein Name | Release Time | Protein Name | Release Time | Protein Name | Release Time |
|---|---|---|---|---|---|
| T1104-D1 | 2022-07-11 | T1106s1-D1 | 2022-06-06 | T1106s2-D1 | 2022-06-06 |
| T1109-D1 | 2022-05-31 | T1119-D1 | 2022-06-08 | T1120-D1 | 2022-07-14 |
| T1120-D2 | 2022-07-14 | T1121-D1 | 2022-06-08 | T1121-D2 | 2022-06-08 |
| T1123-D1 | 2022-05-19 | T1124-D1 | 2022-07-16 | T1129s2-D1 | 2022-07-05 |
| T1133-D1 | 2022-07-18 | T1137s1-D1 | 2022-06-22 | T1137s1-D2 | 2022-06-22 |
| T1137s2-D1 | 2022-05-30 | T1137s2-D2 | 2022-05-30 | T1137s3-D1 | 2022-05-31 |
| T1137s3-D2 | 2022-06-22 | T1137s4-D1 | 2022-06-22 | T1137s4-D2 | 2022-06-22 |
| T1137s4-D3 | 2022-06-22 | T1137s5-D1 | 2022-06-22 | T1137s5-D2 | 2022-06-22 |
| T1137s6-D1 | 2022-06-22 | T1137s6-D2 | 2022-06-22 | T1137s7-D1 | 2022-06-01 |
| T1137s8-D1 | 2022-06-01 | T1137s9-D1 | 2022-06-01 | T1139-D1 | 2022-06-01 |
| T1145-D1 | 2022-08-18 | T1145-D2 | 2022-08-18 | T1150-D1 | 2022-06-11 |
| T1157s1-D1 | 2022-09-01 | T1157s1-D2 | 2022-09-01 | T1157s1-D3 | 2022-09-01 |
| T1157s2-D1 | 2022-09-01 | T1157s2-D2 | 2022-09-01 | T1157s2-D3 | 2022-09-01 |
| T1170-D1 | 2022-07-28 | T1170-D2 | 2022-07-28 | T1180-D1 | 2022-08-24 |
| T1187-D1 | 2022-08-05 | T1188-D1 | 2022-08-05 | T1194-D1 | 2022-08-05 |

Table 10: CASP16 protein names and release times.

| Protein Name | Release Time | Protein Name | Release Time | Protein Name | Release Time |
|---|---|---|---|---|---|
| T1237 | 2024-07-20 | T1206 | 2024-07-17 | T1234-D1 | 2024-09-09 |
| T1276 | 2024-08-19 | T1276-D1 | 2024-08-19 | T1272s3 | 2024-07-29 |
| T1279-D1 | 2024-07-25 | T1214v1 | 2024-08-12 | T1272s9 | 2024-11-14 |
| T1212 | 2024-07-04 | T1272s1 | 2024-07-29 | T1259 | 2024-07-25 |
| T1235 | 2024-07-04 | T1279 | 2024-07-20 | T1298-D1 | 2024-08-16 |
| T1274-D1 | 2024-09-03 | T1201-D1 | 2024-09-24 | T1298-D2 | 2024-08-16 |
| T1272s6-D1 | 2024-11-14 | T1240-D1 | 2024-08-02 | T1201 | 2024-05-19 |
| T1212-D1 | 2024-07-04 | T1237-D1 | 2024-07-20 | T1266-D1 | 2024-07-02 |
| T1235-D1 | 2024-09-09 | T1234 | 2024-07-04 | T1299-D1 | 2024-09-11 |
| T1266 | 2024-07-02 | T1272s7 | 2024-07-29 | T1272s8-D1 | 2024-11-14 |
| T1214v2 | 2024-08-12 | T1272s2-D1 | 2024-08-03 | T1299 | 2024-09-11 |
| T1272s8 | 2024-11-14 | T1274 | 2024-08-14 | T1298 | 2024-08-16 |
| T1272s6 | 2024-11-14 | T1240 | 2024-08-01 | T1210 | 2024-07-15 |
| T1272s2 | 2024-11-14 | T1214-D1 | 2024-09-12 | T1272s4 | 2024-07-29 |
| T1206-D1 | 2024-07-17 | T1259-D1 | 2024-07-25 | T1210-D1 | 2024-09-24 |
| T1272s9-D1 | 2024-11-14 | T1240-D2 | 2024-08-02 | T1279-D2 | 2024-07-25 |
| T1272s5 | 2024-07-29 | T1214 | 2024-07-10 | | |

## A.4 ADDITIONAL FOLDING RESULTS WITH ALPHAFOLD3

To further validate the biological plausibility of DualFold-generated sequences, we extended our AlphaFold3 (Abramson et al., 2024) analysis to four additional CASP16 targets: T1214(length=652), T1235(length=114), T1299(length=68), and T1259(length=204). For each case, we compared the AlphaFold3-predicted structure of the sequence produced by DualFold against the native target backbone.

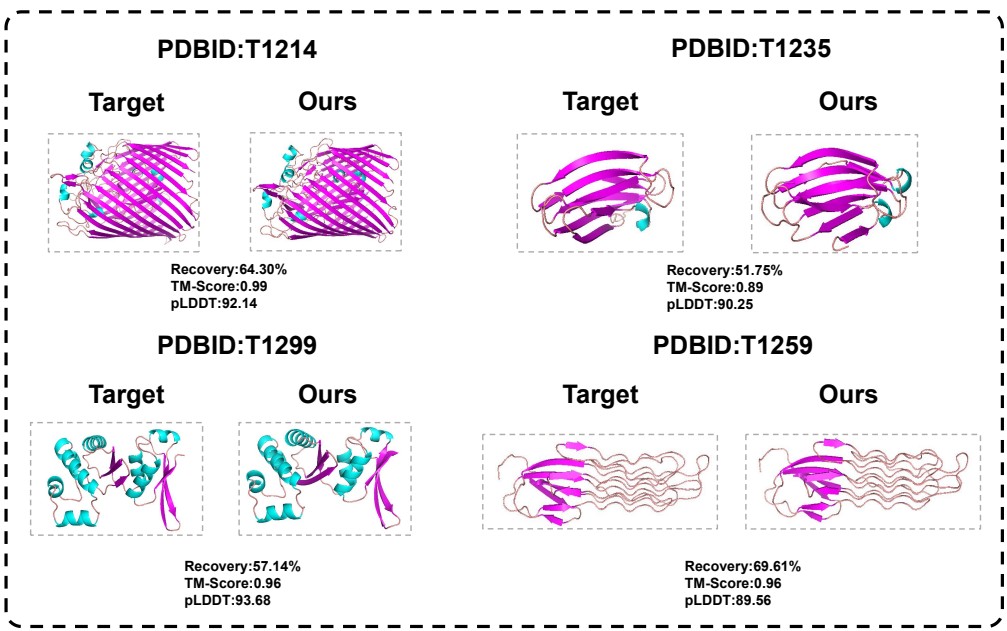

Figure 4: AlphaFold3-predicted structures for sequences designed by DualFold on additional CASP16 targets (T1214, T1235, T1299, and T1259). Each row shows an overlay of the target structure and the predicted structure from the DualFold-designed sequence. Across all cases, the strong alignment provides further support for the biological plausibility of DualFold outputs.

As illustrated in Figure 4, the predicted structures of all four designed sequences align closely with their respective native folds. Visually, the confidence levels (pLDDT) remain consistently high, and the structural overlays show strong backbone agreement. These results further reinforce that DualFold is capable of generating sequences with not only improved design metrics but also high in silico structural plausibility across targets of different lengths and topologies.

## A.5 LoRA CONFIGURATION FOR MPLM AND PLM FINE-TUNING

Both models employ identical LoRA hyperparameters to ensure balanced adaptation. We use a rank of 8, which provides a good balance between parameter efficiency and model expressivity. The scaling factor is set to 32, offering sufficient learning capacity for the inverse folding task while maintaining stability. A dropout rate of 0.1 is applied to the LoRA layers for regularization, and we disable bias adaptation to further reduce the parameter count.

This configuration results in approximately 0.1% trainable parameters relative to the full model size, significantly reducing computational overhead while maintaining the models' ability to adapt to the inverse folding task. The low rank ensures efficient adaptation, while the relatively high scaling factor compensates for the reduced parameter space, allowing the models to learn task-specific patterns effectively.

The LoRA adaptation is implemented using the Parameter Efficient Fine-Tuning (PEFT) framework. During initialization, we load the pretrained ESM-3 and ESM-C models, configure the LoRA adapters with the specified hyperparameters, and freeze all base model parameters. Only the LoRA weights are updated during training, which prevents catastrophic forgetting of the pretrained knowledge while allowing task-specific adaptation.

The symmetric configuration across both models ensures balanced adaptation of sequence and structural knowledge during training. By targeting both attention and feed-forward components, we enable the models to adapt their representation learning capabilities while preserving the fundamental understanding encoded in the pretrained weights.

## A.6 MATHEMATICAL FORMULATIONS FOR EVALUATION METRICS

### Perplexity

Perplexity is a fundamental metric for evaluating the quality of language model predictions, defined as the exponential of cross-entropy loss. In protein sequence design tasks, we utilize perplexity to assess the model's accuracy in predicting authentic amino acid sequences.

Given a protein sequence $\mathbf{s} = \{s_1, s_2, \ldots, s_\ell\}$, where $s_i$ represents the amino acid at position $i$ and $\ell$ is the sequence length. The model outputs logits denoted as $\mathbf{z} = \{\mathbf{z}_1, \mathbf{z}_2, \ldots, \mathbf{z}_\ell\}$, where $\mathbf{z}_i \in \mathbb{R}^{|\mathcal{V}|}$ and $|\mathcal{V}|$ is the vocabulary size (20 standard amino acids).

First, we compute the log probability distribution at each position:

$$p_i(k) = \frac{\exp(z_{i,k})}{\sum_{j=1}^{|\mathcal{V}|} \exp(z_{i,j})} \tag{7}$$

where $z_{i,k}$ represents the logit value for predicting amino acid $k$ at position $i$.

The log-likelihood for the target amino acid is:

$$\log p_i(s_i) = \log \left( \frac{\exp(z_{i,s_i})}{\sum_{j=1}^{|\mathcal{V}|} \exp(z_{i,j})} \right) = z_{i,s_i} - \log \sum_{j=1}^{|\mathcal{V}|} \exp(z_{i,j}) \tag{8}$$

### Global Perplexity Calculation

Considering the presence of special tokens such as padding, class (cls), and end-of-sequence (eos) tokens in protein sequences, we define a valid position mask:

$$\mathcal{M}_i = \mathbb{K}[s_i \neq \text{pad}] \cdot \mathbb{K}[s_i \neq \text{cls}] \cdot \mathbb{K}[s_i \neq \text{eos}] \cdot c_i \tag{9}$$

where $c_i$ is the coordinate mask used to identify structurally valid positions, and $\mathbb{K}[\cdot]$ is the indicator function.

The global average negative log-likelihood is:

$$\text{NLL}_{\text{global}} = -\frac{\sum_{i=1}^{\ell} \mathcal{M}_i \cdot \log p_i(s_i)}{\sum_{i=1}^{\ell} \mathcal{M}_i} \tag{10}$$

The final global perplexity is defined as:

$$\text{PPL}_{\text{global}} = \exp(\text{NLL}_{\text{global}}) \tag{11}$$

### Sequence Recovery Rate

The sequence recovery rate measures the degree of matching between generated sequences and target sequences at the amino acid level, serving as a direct indicator for evaluating the accuracy of protein sequence design.

Given a predicted sequence $\hat{\mathbf{s}} = \{\hat{s}_1, \hat{s}_2, \ldots, \hat{s}_\ell\}$ and a target sequence $\mathbf{s} = \{s_1, s_2, \ldots, s_\ell\}$, the sequence recovery rate is defined as the proportion of correct predictions at valid positions.

The correctness indicator function for individual positions:

$$\delta_i = \mathbb{K}[\hat{s}_i = s_i] \cdot \mathcal{M}_i \tag{12}$$

where $\mathcal{M}_i$ is the valid position mask defined previously.

### Sequence-level Recovery Rate

For a single sequence, the recovery rate is calculated as:

$$\text{Recovery}_{\text{seq}} = \frac{\sum_{i=1}^{\ell} \delta_i}{\sum_{i=1}^{\ell} \mathcal{M}_i} \tag{13}$$

**Dataset-level Statistics**

For a test set containing $N$ sequences, we compute the recovery rate for each sequence and then calculate the median as the final evaluation metric:

$$\text{Recovery}_{\text{median}} = \text{median}\{\text{Recovery}_{\text{seq}}^{(1)}, \text{Recovery}_{\text{seq}}^{(2)}, \ldots, \text{Recovery}_{\text{seq}}^{(N)}\} \tag{14}$$

The use of median instead of mean reduces the impact of outliers and better reflects the model's overall performance.

### A.7 THE USE OF LLMs

This work employed large language models (LLMs) as a general-purpose assistance tool, primarily for polishing initial drafts of non-core sections of the manuscript. LLMs were not involved in research conception, methodological development, experimental design, or data analysis. The authors take full responsibility for all content, including any LLM-generated material, and confirm that no plagiarism or scientific misconduct is present.

