# OpenReview forum: "Symmetric Dual-Path Integration for Protein Inverse Folding"
_ICLR.cc/2026/Conference — ICLR 2026 Conference Withdrawn Submission_

### Official Review · Reviewer_xDMk · 2025-10-26

**Soundness:** 2
**Presentation:** 2
**Contribution:** 1
**Rating:** 4
**Confidence:** 3

**Summary:**

The manuscript proposes DualFold, a protein inverse folding framework that integrates a pretrained protein language model and a multimodal protein language model. DualFold employs a structure encoder and a fine-tuning module to enhance sequence recovery accuracy. Experimental results show that DualFold outperforms baseline methods on several benchmark datasets.

**Strengths:**

- The manuscript is clearly written and well-organized.
- The method achieves superior performance compared to existing baselines in the experiments presented.

**Weaknesses:**

- The method appears to be a relatively simple combination of a PLM and an MPLM, without introducing substantial architectural innovations.
- Although the performance gains are notable, there are several concerns regarding experimental fairness and interpretation. For example, for results in Table 4, it’s unclear whether the numbers of trainable parameters are controlled or not.
- The experiments are conducted using only one type of PLM and MPLM, leaving it unclear how the method performs with alternative model choices.

**Questions:**

1. In Table 4, when evaluating the complementary roles of PLM and MPLM, are the numbers of trainable parameters controlled to be the same across variants (i.e., without PLM or without MPLM)? If not, the comparison may not be entirely fair.
2. The ablation study shows that removing the PLM leads to a larger performance drop than removing the MPLM. This seems counterintuitive, since the MPLM should, in principle, already incorporate the knowledge from the PLM. Could the authors provide an explanation for this observation?
3. How does the performance change when using different PLMs and MPLMs?

---

### Official Review · Reviewer_igq6 · 2025-10-28

**Soundness:** 2
**Presentation:** 3
**Contribution:** 2
**Rating:** 4
**Confidence:** 4

**Summary:**

This paper studies the important inverse folding problem in protein design. This paper introduces DualFold a harmonic dual-path architecture that both leverages PLMs for pretrained sequence knowledge and MPLMs for pretrained structural knowledge to iteratively guide protein sequence generation. DualFold achieves state-of-the-art performance in the CATH, TS, and CASP datasets.

**Strengths:**

- This paper is well-written and easy to follow.
- The authors conduct extensive ablation studies to analyze the design choice.

**Weaknesses:**

- In line 124, the authors said that LM-Design, KW-Design, and Bridge-IF lack access to structural constraints. However, all three models introduce another structure module to capture the structural conditions in the refinement process.
- The overall architecture is similar to KW-Design. The better performance may result from the better PLM (ESM-C) and MPLM (ESM-3). It is necessary to compare the aforementioned baselines with the sample PLM or MPLM.
- ProteinMPNN is widely used in de novo protein design when combined with RfDiffusion[1] and AF3. It is interesting to evaluate whether the proposed method can be used in such an important case [1][2][3].

[1] De novo design of protein structure and function with RFdiffusion

[2] Learning Inverse Protein Folding with Markov Bridges

[3] A Holistic Evaluation of Protein Foundation Models

**Questions:**

Please see Weaknesses

---

### Official Review · Reviewer_qS8d · 2025-10-31

**Soundness:** 2
**Presentation:** 3
**Contribution:** 2
**Rating:** 4
**Confidence:** 3

**Summary:**

This paper presents DualFold, a dual-path architecture for protein inverse folding that integrates a protein language model (PLM) and a multimodal protein language model (MPLM). The method aims to leverage both sequence and structural priors through an adaptive fusion mechanism and a self-correction training strategy.
The framework uses ESM-C (600 M) as the PLM and ESM-3 1.4 B as the MPLM backbone, both frozen with LoRA adapters. DualFold is evaluated on CATH 4.2, TS50/TS500, and CASP 15/16, showing consistent improvements in recovery and perplexity. Ablation results highlight the contributions of both paths and the two-stage self-correction.

**Strengths:**

1. The paper is technically solid and inspiring. The integration of PLM and MPLM is intuitive and implemented cleanly.
2. Evaluations on multiple benchmarks and detailed ablations (w/o PLM, w/o MPLM, w/o self-correction) make the empirical evidence convincing.

**Weaknesses:**

1. The “symmetric” claim is misleading: the fusion (Eq. 4) is directional; MPLM outputs act as weighted residuals on PLM predictions without reciprocal information flow.
2. The statement that “MPLMs alone are insufficient for inverse folding” appears somewhat overclaimed, since the authors evaluate only a single MPLM implementation in a zero-shot setting, without task-specific finetuning or comparison to other multimodal baselines.

**Questions:**

1. The model initializes generation with PiFold/ProteinMPNN features. What would happen if the initial sequence were randomly initialized instead? Would the dual-path fusion and self-correction mechanisms still perform effectively?
2. The authors use relatively small backbones (ESM-C 600 M and ESM-3 1.4 B). Have you explored whether performance scales with larger PLM/MPLM variants? It would be interesting to see if the dual-path fusion remains necessary when stronger individual models are used.
3. Both the PLM and MPLM backbones (ESM-C and ESM-3) were pretrained on large protein corpora prior to fine-tuning. It would be helpful if the authors could specify the pretraining datasets and cutoff dates, and clarify whether any CATH or CASP evaluation proteins (or their close homologs) were included in these corpora. Providing this information would help rule out potential data leakage from pretraining into evaluation.

---

### Official Review · Reviewer_gJGg · 2025-11-01

**Soundness:** 1
**Presentation:** 3
**Contribution:** 1
**Rating:** 2
**Confidence:** 5

**Summary:**

The paper proposes DualFold, a dual-branch framework combining a protein language model (PLM) and a multimodal protein language model (MPLM; specifically ESM-3) for protein inverse folding. The two branches (sequence and structure) produce residue-wise logits that are fused via a learned weighting function and jointly fine-tuned with a self-correction iterative training strategy. The authors claim that this symmetric integration improves recovery and perplexity across classical benchmarks (CATH-4.2/4.3, TS50, TS500) and subsets of CASP15/16.

**Strengths:**

- Clean presentation of the designed method.
- Extensive quantitative results on multiple benchmarks (although almost certain data leakage).
- Attempts to evaluate on more recent CASP15 and CASP16 targets.
- Empirical analysis (ablation, iterative inference) is clear.

**Weaknesses:**

$\textbf{1. Fundamental flaw: Testing on training dataset:}$

- The paper uses ESM-3 as their MPLM, which has been pre-trained on all the sequence and structure pairs in PDB, AlphaFold-DB, and ESMAtlas (ref: [1], Appendix Sec. A.2.1.3, and A.2.1.5.).

  - During training, ESM-3 has seen all the PDB structures deposited before May 1, 2020 (ref: [1], Appendix A.2.1.5). Now CATH-4.2, CATH-4.3, TS50, TS500 are either subsets or have strong overlap with these PDB entries (please see [2,3,4,5]). ESM-3, being a multimodal PLM trained to learn the joint distribution (of structure, sequence, and function), has already been trained to map structure-sequence of these test sets. So evaluating on these test sets is basically $\textbf{evaluating on subsets of the (pre-)training set as is}$, which is a fundamental issue.

  - Not only ESM-3 has very likely been trained on these samples, but also they have augmented sequence data for each of the structures with an inverse folding model that has been trained with sequence-structure pairs in PDB, AlphaFold-DB, and ESMAtlas (ref: [1], Appendix A.2.1.3).

- While both MPLM (ESM-3) and PLM (ESM-C) has been trained on data that are present in these test sets, MPLM creates a more concerning case due to the fact that ESM-3 has directly been trained to map structure-sequence, while ESM-C has been trained to learn the distribution of the sequences only (not the direct task of inverse folding). Now, since the authors directly add ESM-3’s logits (weighted sum) with the PLM’s logits to produce the output logits (Eq. 4), it very likely becomes more of a recall from memory.

$\textbf{2. Minimal improvement on TS500 despite likely been seen during pre-training:}$

- Even though TS500 is likely also present in the MPLM’s pre-training set, DualFold’s performance improvement over KW-Design is minimal (70.48 vs 69.19) compared to other dataset such as CATH-4.2, CATH-4.3, and TS50. This raises the question of any actual effect of the proposed design of DualFold.


$\textbf{3. Missing essential baselines on the newer test sets:}$

- The model is evaluated on CASP15 and CASP16, which likely does not have temporal overlap with pretraining datasets (mentioned above) or the finetuning datasets (CATH-4.2, CATH-4.3). This makes these datasets good candidates for true performance evaluation. However:

  - The authors compared only against a $\textit{subset of the previous methods}$; more concerning is that none of these is a more recent method. This makes it impossible to evaluate the true impact of the improvement claimed in this paper.

  - Both DualFold and KW-Design [2] were evaluated on the same 45 samples from CASP15 (checked against target IDs reported in Table 9), the authors of DualFold did not seem to report KW-Design’s performance on this test set, while they compared against KW-Design on CATH-4.2, CATH-4.3, TS50, and TS500. For these 45 samples in CASP15, KW-Design achieved a median recovery of $\textbf{about 56}$% for argmax sampling, which is almost the same as what DualFold achieved ($\textbf{56.57}$%) (ref: [2], Appendix C, Figure 7). This makes us wonder whether DualFold can provide any recovery improvement for proteins outside its pre-training set.

$\textbf{4. Method lacks true novelty}$

- The term “symmetric dual-path integration” implies architectural innovation, but the proposed approach is practically an ensemble with late-stage logit fusion (equivalent to a trainable ensemble weight), which is rather incremental.

- As we can see in Eq. 4, it practically ensembles the output logits form 3 models:

  - From the structure encoder such as ProteinMPNN or PiFold ($f^*_e$), from the PLM ($f_s$), from the MPLM ($f_m$).

  - The final logits $f_{\text{dual}}(f_s, \hat{\mathbf{s}}, f_m, \mathbf{x})_i = f_s(\hat{\mathbf{s}})_i + (\mathbf{w}^\top f_e^*(\mathbf{x})) \cdot f_m(\mathbf{x}, \hat{\mathbf{s}})_i$ is practically an element-wise multiplication of logits from two predictors and added with the logits of the third.

- Existing methods such as DPLM [4], LM-Design [5], already fuse two paths through an adaptor (fusing sequence embedding and structure embedding).

  - Moreover, LM-Design showed that fusing the logits of structure encoder with the adapter’s logits already significantly improves performance (ref: [5], Appendix Sec. D.2 and Tab. 6).

  - I would suggest authors clarify on their methodological innovation, including how their proposed approach is different from these existing ones.


$\textbf{4. Lack of foldability experiments}$

- The authors provide foldability scores on a few samples (6 only), rather than a more standard (re-)foldability evaluation on one or more entire test sets, ideally CASP15 and/or CASP16, and comparison against other state-of-the-art methods. While I appreciate the interesting results on proteins of different lengths, I would recommend the authors show proper foldability comparison against at least 1 or 2 $\textit{recent}$ inverse folding methods on CASP15 and/or CASP16 for a clearer comparison of the performance.

$\textbf{5. Missing computational complexity and running time analysis}$

- Since the approach uses two large PLMs at each refinement step, it would likely create some computational overhead (and/or likely additional running time). Especially inference time per protein is an essential metric for a protein design or inverse folding method for it to be practically useful. I would suggest the authors analyze the computational overhead and running time (wall-clock time) analysis, comparing against the structure encoder-only, PLM-only, MPLM-only variants.

$\textbf{6. Claim issue}$

- The authors' claim that they "introduce MPLMs into the inverse folding" is not true. Some existing methods (e.g., DPLM2 [6]) already evaluated inverse folding performance with their proposed MPLMs.


[1] Hayes et al., Simulating 500 million years of evolution with a language model, Science 2025, https://doi.org/10.1101/2024.07.01.600583


[2] Gao et al., Knowledge-Design: Pushing the Limit of Protein Design via Knowledge Refinement, ICLR 2024, https://openreview.net/pdf/907911cdd7b1bf8e10eee57bbfed18dcd923e03d.pdf

[3] Hsu et al., Learning inverse folding from millions of predicted structures, ICML 2022, https://proceedings.mlr.press/v162/hsu22a/hsu22a.pdf

[4] Wang et al., Diffusion Language Models Are Versatile Protein Learners, ICML 2024, https://openreview.net/pdf?id=NUAbSFqyqb

[5] Zheng et al., Structure-informed Language Models Are Protein Designers, ICML 2023, https://proceedings.mlr.press/v202/zheng23a/zheng23a.pdf

[6] Wang et al., DPLM-2: A Multimodal Diffusion Protein Language Model, ICLR 2025, https://openreview.net/pdf/13c6bc9b904922e7352e690eae7cad8a2d4526f7.pdf

**Questions:**

1. Can the authors quantify any overlap between their evaluation datasets (CATH-4.2/4.3, TS50, TS500, CASP15/16) and ESM-3’s pretraining data?
2. How is the proposed dual-path fusion mathematically or functionally different from prior adaptor-based models like DPLM or LM-Design, and simple logit-fusion?
3. Can the authors provide full-set foldability evaluations (e.g., CASP15/16) instead of six selected samples?
4. Can the authors kindly add comparison against recent methods on CASP15/16; or justify why the recent baselines should be excluded from comparison, especially those that have already reported scores on these dataset in their papers?

---

### Note · Authors · 2025-11-15

**Comment:**

I have read and agree with the venue's withdrawal policy on behalf of myself and my co-authors.

**Withdrawal Confirmation:**

I have read and agree with the venue's withdrawal policy on behalf of myself and my co-authors.